# Structural and Functional Insights into the Microtubule Organizing Centers of *Toxoplasma gondii* and *Plasmodium* spp.

**DOI:** 10.3390/microorganisms9122503

**Published:** 2021-12-03

**Authors:** Ramiro Tomasina, Fabiana C. González, Maria E. Francia

**Affiliations:** 1Laboratory of Apicomplexan Biology, Institut Pasteur de Montevideo, Montevideo 11400, Uruguay; rtomasina@pasteur.edu.uy (R.T.); fcgonzalez@pasteur.edu.uy (F.C.G.); 2Departamento de Parasitología y Micología, Facultad de Medicina, Universidad de la República, Montevideo 11600, Uruguay

**Keywords:** microtubule organizing center, centrosome, centriolar plaque, *Plasmodium*, *Toxoplasma gondii*, ultrastructure expansion microscopy

## Abstract

Microtubule organizing centers (MTOCs) perform critical cellular tasks by nucleating, stabilizing, and anchoring microtubule’s minus ends. These capacities impact tremendously a wide array of cellular functions ranging from ascribing cell shape to orchestrating cell division and generating motile structures, among others. The phylum Apicomplexa comprises over 6000 single-celled obligate intracellular parasitic species. Many of the apicomplexan are well known pathogens such as *Toxoplasma gondii* and the *Plasmodium* species, causative agents of toxoplasmosis and malaria, respectively. Microtubule organization in these parasites is critical for organizing the cortical cytoskeleton, enabling host cell penetration and the positioning of large organelles, driving cell division and directing the formation of flagella in sexual life stages. Apicomplexans are a prime example of MTOC diversity displaying multiple functional and structural MTOCs combinations within a single species. This diversity can only be fully understood in light of each organism’s specific MT nucleation requirements and their evolutionary history. Insight into apicomplexan MTOCs had traditionally been limited to classical ultrastructural work by transmission electron microscopy. However, in the past few years, a large body of molecular insight has emerged. In this work we describe the latest insights into nuclear MTOC biology in two major human and animal disease causing Apicomplexans: *Toxoplasma gondii* and *Plasmodium* spp.

## 1. Introduction

Invariably, microtubule organizing centers (MTOCs) perform critical cellular tasks by nucleating, stabilizing, and anchoring microtubule’s (MT) minus ends. These capacities are required for a wide array of functions, including ascribing cells with their characteristic shape and polarity, influencing resistance to mechanical forces, organizing the mitotic spindle, or acting as basal bodies (BBs) positioning and contributing to the nucleation and motility of cilia and flagella, and in this way, impacting intracellular transport, signaling, and cellular differentiation. The fascinating biology of MTOCs, and their role in cellular life, health, and disease has attracted scientists since the late 1800s [1]. A remarkable landmark in MTOC research history is Theodore Boveri’s monograph ‘Ueber die Natur der Centrosomen’ written in the 1900s. His initial observations of mitotically dividing fertilized eggs of the nematode *Parascaris equorum* (then *Ascaris megalocephala*) established that the formation of mitotic spindles was controlled by a cellular organelle which persisted, replicated, and was inherited. His coining of the term “centrosome” in 1887 [2] sparked the interest for this structure which has provided, since, over 120 years of wonderland for microscopists and cell biologists.

In the best studied higher eukaryote systems—such as humans and other animals—the centrosome performs the majority of the MTOC aforementioned functions. Animal centrosomes are characterized by the presence of two MT-based barrels known as centrioles. Centrioles are precisely organized. Centriolar MT organize following a nine-fold radial symmetry determined by a foundational structure known as the cartwheel. The cartwheel is a structure formed onto a pre-existing centriole (a “mother” centriole), guided by the self-assembling properties of the Spindle assembly abnormal protein 6 homolog (SAS6) [3,4,5,6,7,8,9]. SAS6 interacts with other proteins, such as SCL-interrupting locus protein (STIL) and CEP135 to complete the cartwheel structure defining the geometric arrangement of the nine MT triplets. These triplets are each made up of a complete MT (A-tubule) and two incomplete tubules (named B and C). Precisely controlled MT polymerization completes the centriole cylindrical structure. The length of the centriolar MTs growing apically from the cartwheel is tightly regulated and cell-type specific [10].

MT nucleation by the human centrosome occurs at all times. During interphase, the centrosome organizes the cell’s cortical cytoskeleton. During cell division, MT organization is focused on the mitotic spindle. A complex matrix of proteins, known as the pericentriolar material (PCM), are orderly layered onto the centrioles. The PCM ascribes centrioles with their capacity for MT nucleation (reviewed in [11]). The PCM’s ability to nucleate MTs is supported on multiple proteins out of which gamma-tubulin, the gamma-tubulin ring complex, and their transient interactors along the cell cycle, play pivotal roles in catalyzing and regulating the MT nucleation process [12].

It should be noted that in spite of the human centrosomes being by far the most intensely studied and the best understood both structurally and functionally (largely owed to their preponderant role in underlying genetic conditions such as infertility, cancer, and ciliopathies), their structure and organization is by no means the most representative found in nature. In fact, MT nucleation does not require centrosomes bearing bona fide centrioles; it can occur centriole-independent even in humans [13].

The phylum Apicomplexa comprises over 6000 single-celled parasitic protozoan species. Many of the apicomplexan are well known pathogens, causing immense morbidity and mortality both in humans and animals. *Toxoplasma gondii*, *Plasmodium* spp., and *Cryptosporidium* spp., are causative agents of toxoplasmosis, malaria, and cryptosporidiosis, respectively. Invariably all species within the phylum are obligate intracellular parasites, exhibiting multiple life stages—all happening within the host cells of definitive and intermediate hosts.

MTs organization in apicomplexan parasites encompasses organizing the cortical cytoskeleton, required to enable host cell penetration and the positioning of large organelles [14,15], as well as cell division and the formation of flagella in certain life stages.

Intracellular life is attained by active invasion. Active invasion implies the formation of a tight junction, which literally describes the intimate contact formed between the host cell plasma membrane and the parasite membrane. The tight junction constricts the parasite as it enters the cell, closing behind the parasite as invasion completes, thus avoiding the lysis of the infected cell. This invasion mechanism requires a robust cytoskeleton capable of penetrating the cell and withstanding mechanical forces experienced by the parasite as it enters. Proper cytoskeleton assembly is also required for motility. Apicomplexa have achieved such cortical MT stability by organizing a corset of subpellicular MTs which twirl around the cell body and extending through two thirds of the cell length. In conjunction with a specialized membranous system, this corset provides the required mechanical resistance as well as flexibility, allowing these parasites to invade our cells.

The subpellicular MTs in *T. gondii* and *Plasmodium* species are organized by ring-shaped MTOCs located at the cell apex, known as the apical polar ring (APR) [16,17,18,19,20,21,22,23,24]. In addition to the complexity of cortical MT organization during asexual development, the apicomplexan parasites also undergo sexual differentiation into macro and microgametes. While asexual stages are aflagellated, microgametes formed during sexual differentiation display flagella; mature *Plasmodium* sperm have a single flagellum organized by a centriole-like structure composed of nine single A-type tubules (complete MTs) and no central tube, embedded in an electron-dense mass. *T. gondii* microgametes, on the other hand, display two flagella, and the structure of their originating BB remains debatable [25,26].

Whether BBs derive from a repurposed MTOC previously present in the asexual stages or distinct MTOCs are assembled de novo, is also unclear. However, recent data has shed light onto the process in *Plasmodium*. *Plasmodium* simultaneously forms eight flagellated sperm cells [25,27,28]. Isotropic parasite expansion combined with whole proteome labeling (Pan-UExM) allowed visualization of the BB formation kinetics with unprecedented temporal resolution. BBs were shown to simultaneously form from a deuterostome-like structure (an MTOC originating proteinaceous matrix devoid of centrioles). In addition, a proteinaceous matrix was found linking the newly formed BBs with the nuclear MTOC, thereby physically linking both MTOCs in these cells [29]. Partly owed to the life stage’s experimental inaccessibility in the cat’s gut, the route of BB biogenesis in the *T. gondii* male gametes is poorly understood. However, new technical breakthroughs allowing microgamete formation in vitro will likely expedite our understanding of the process in the coming years [26,30,31].

Apicomplexa divide by divergent mechanisms (recently reviewed in [32]) (Figure 1A,B). A diversity of sexual and asexual cell division modes are used by these parasites to proliferate. With a few notable exceptions within the phylum (e.g., *Babesia*—dividing by binary fission; *Theileria* spp.—dividing by hitchhiking on the host cell’s division apparatus) apicomplexan parasites can follow three division modes: endopolygeny, endodyogeny, and schizogony. These modes vary in the extent to which chromosome replication is followed by nuclear mitosis and cytokinesis. In endopolygeny, chromosomes are replicated several times before the nucleus undergoes mitosis. Mitosis is then followed by parceling of multiple nuclei simultaneously into tens of daughter cells. In endodyogeny, each nuclear division cycle (encompassing DNA replication and nuclear mitosis) is followed by daughter cell formation and cytokinesis (Figure 1A). Finally, in schizogony, nuclei undergo asynchronous DNA synthesis and mitosis, followed by a final round of synchronized mitoses coordinated with the simultaneous formation of several dozen daughter cells (Figure 1B). Remarkably, *T. gondii* divides by all three mechanisms as it transitions through its different life forms in different hosts (reviewed in [26,32,33,34]).

Each of these three modes of division encompass specific MT nucleation requirements. However, all modes pose similar challenging settings for MT nucleation from a standpoint of topological constraints. In all cases, chromosome segregation occurs by semi-closed mitosis, in the presence of a visibly unchanged nuclear envelope and indetectable chromatin condensation [35,36,37]. In addition, nuclear division occurs at one point or another in synchrony with daughter cell cortical microtubule cytoskeleton formation. The latter occurs de novo—at the mother cell surface in schizogony or at the mother cell cytosol in endopolygeny and endodyogeny [32]. The MTOCs for each forming daughter cell (i.e., an APR) must be precisely positioned and in concordance with the number of full chromosomal complements present at the mother cell at the time of daughter cell formation. Apicomplexans have solved this conundrum by separately controlling nuclear events and daughter cell formation evolving two functionally related and physically connected, albeit distinct, microtubule organizing centers [38].

Apicomplexans are a prime example of MTOC diversity and multiplicity of functional combinations within a single species. In this work we describe the latest insights into nuclear MTOC biology in two major human and animal disease causing Apicomplexans: *T. gondii* and *Plasmodium* spp.

## 2. Nuclear Division Organization by MTOCS of *T. gondii* and *Plasmodium*: Structural and Functional Insight

Centrosomal architecture in Apicomplexan parasites is highly diverse and has been repeatedly referred to as being highly divergent. The latter holds true in reference to the centrosomal architecture of animals. Notable differences exist even amongst the centrosomes of *T. gondii* and the *Plasmodium* spp.; the most outstanding one being that *T. gondii* bears MT-barrel based centrioles while *Plasmodium* species do not.

Schizogonic cell division in blood-stage *Plasmodium* encompasses asynchronous mitoses of multiple co-existing nuclei in a shared mother cell cytosol. Each individual nucleus controls its own MT nucleation requirements by bearing its own MTOC. The nuclear MTOC in *Plasmodium* was originally named the “centriolar plaque” (CP) and is also referred to in the literature as the “kinetic center”. The CP was originally identified as an electron dense focus, proposed to be embedded in the nuclear envelope, and in close connection to a nuclear pore [27,35,39]. The CP was later shown to be home to many bona fide centrosomal proteins including centrin and gamma-tubulin, and to house MT nucleation capacity [40,41].

During asexual development, blood-stage *Plasmodium* nuclei undergo dynamic changes of their nuclear MTs. Non-dividing nuclei bear a single CP, and intranuclear MTs. The latter form a “hemi-spindle” composed of a handful (~5) of bundled individual MTs. The hemi-spindle extends from the single CP to the opposite side of the nucleus. Upon the onset of S-phase, or DNA replication, the hemi-spindle retracts. This is followed by CP duplication and the formation of a mitotic spindle. At this point, the spindle presumably connects to the chromosome’s kinetochores, but also keeps the CPs interconnected by extended MTs spanning the nucleus. An illustrative transmitted electron micrograph of the spindle at this stage is shown in [35]. More recently, this spindle has been visualized by fluorescence microscopy, and the term “interpolar” spindle coined [42].

Early transmission electron microscopy work in *P. berghei* looking into sporogony—the mode of division used by the sporozoites life stage, present at the mosquitoes’ salivary glands—defined the interpolar spindle as consisting of three distinct MT populations: the MTs spanning the duplicated CPs, the ones contacting the kinetochores, and MTs extending from one CP to a point beyond the equatorial plane of the spindle of unknown function [43]. As sister chromatids separate, and karyokinesis advances, the interpolar spindle retracts, and nuclear fission occurs.

The advent of immunofluorescence, in the early 90s, allowed further resolving the dynamic changes undergone by MTs in *P. falciparum*. Using anti-tubulin antibodies, it was shown that while the majority of spindles exist in 180° configurations, the interpolar spindle can exist in a variety of other configurations which are nonetheless productive [22]. Spindle elongation was observable as chromatids separated from each other. It was also shown that the spindle evolves into a hemi-spindle upon sister chromatid separation, followed by an accumulation of nuclear diffuse tubulin staining. This suggests that the spindle-derived hemi-spindle disassembles following mitosis, and that no mitotic-derived spindle persists beyond mitosis [22].

More recently, ultrastructure expansion microscopy (UExM) in *P. falciparum* has allowed a (literal) closer look into MT dynamics using fluorescent markers. In addition to providing an approximate four-fold increase in resolution, the co-staining of MTs and a membrane stain (BODIPY), allowed for the first time the concomitant visualization of MT and nuclear envelope dynamics [42]. Concurrent observation of both MT and the NE is critical to understanding each structure’s role in DNA segregation during closed mitosis.

The spindle pole body (SPB) of yeast is a multilayered structure either embedded in the nuclear envelope (in the budding yeast *Saccharomyces cerevisiae*) or inserted into the nuclear envelope prior to mitosis (fission yeast, e.g., *Schizosaccharomyces pombe*). In either case, the SPB is not only responsible for nuclear MT nucleation during mitosis, but also organizes the cortical cytoskeleton. Like the CP in *Plasmodium*, SPBs are devoid of centrioles. Given its localization, apposed to the nuclear envelope, and the lack of *bona fide* centrioles, the CP has often been modeled after the yeast SPB. However, recent insights from UExM have revealed that the CP is in fact extranuclear, and is not embedded in the nuclear envelope at any point during mitosis. UExM, stimulated emission depletion (STED) microscopy, and correlative light electron microscopy (CLEM) has further revealed that nucleation of microtubules resides within a chromatin free sub-compartment within the nucleus, distal to the location of Centrin, a CP component, and proximal to a nuclear pore marker (Nup313) [44]. This study has clearly established that whereas the CP is at the poles of the mitotic spindle, the CP is extranuclear, and MT nucleation capacity resides at a protein-dense region internal to the nucleus, but devoid of chromatin (Figure 1D).

Importantly, centromeres—the chromosomes’ regions onto which the kinetochore assembles—have been precisely mapped in *P. falciparum*. The histone variants PfCENH3, the prime molecular marker of centromeres, and PfH2A.Z, occupy a 4–4.5 kb region of similar size and sequence composition in all *P. falciparum* chromosomes. Immunofluorescence assay of PfCENH3 revealed that centromeres undergo dynamic changes in localization during division. In stages ranging from early trophozoites to mature schizonts, centromeres cluster to a single nuclear location proximal to the CP prior to and during mitosis and cytokinesis. However, centromeres dissociate soon after invasion in ring stages, whereby multiple PfCENH3 foci per nucleus are observed [45]. Strikingly, however, the centromere clustering localization during mitosis does not coincide with the central region of the spindle. This region of the spindle has been regarded as a metaphase plate. Instead, centromeres seem to cluster at the base of the chromatin-free region from which MTs are nucleated during mitosis (see in [44]). This implies that either there is a short spindle at the site of centromere clustering—undetectable by various microscopy techniques—or that centromeres could be segregated by connecting to the nuclear envelope. Finally, it is possible that regions other than the centromeric chromatin bear physical connections to the nuclear envelope or the various spindles assembled throughout the life of the nucleus. Combining UExM, with centromere and MT markers, as well as BODIPY to visualize the nuclear envelope, should clarify this matter.

Much of our understanding of MTOC biology has come from analyzing their structural and functional variability in light of what has been described in humans or in other better studied model species such as yeast. Many studies have focused efforts in describing presence/absence of human centrosomal components in other eukaryotes. However, as mentioned above, these MTOCs represent only a minute fraction of the diversity present in nature.

Contrary to the plethora of information available for the yeast SPB (whose all 18 protein components have been mapped and localized [46]), or the human centrosome (whose proteome has been identified even with spatial resolution [47,48]) information of the molecular make-up of the CP is scarce. To our knowledge, no systematic identification of CP components has been pursued. In fact, only a handful of molecular components have been definitely localized to the CP structure, and even fewer have been functionally validated (Table 1).

*Plasmodium*’s genome seems to lack homologs of many of the well-known centrosomal proteins. Proteins like Spindle and centriole-associated protein 1 (SPICE), CEP192, CEP63, CEP152, Spindle assembly abnormal protein 5 (SAS-5),CP110, Centrobin, POC5, C2CD3, Ofd1, Polo-like kinase -1 (PlK-1), and PlK-4 are not present in the genome of this parasite. Cep135 (also known as Bld10) interacts with SAS6 to assemble the centriolar cartwheel in animals and Drosophila. Interestingly, despite not displaying centrioles, Plasmodium encodes for a homolog of Cep135 (in *P. falciparum*; encoded by PfML01_060030300) and SAS6 [58,59]. It is plausible, however, that instead of forming the CP, these proteins could participate in basal body formation during gametogenesis.

Centrin and Ɣ-tubulin are the prime markers identifying the CP structure [60]. While the latter plays a role in MT nucleation, the former belongs to a family of EF-hand containing Calcium binding proteins, shown to play pivotal roles in centrosome duplication and segregation in other systems [61]. The *Plasmodium* genome encodes for at least four Centrin-related proteins, whereby Centrin1 and Centrin3 are paralogues of mammalian centrins, and Centrin2 and Centrin4 are alveolate-specific [62]. Both Centrin2 and Centrin3 have been shown to localize at the CP by immuno-EM and confocal microscopy [62] in *P. falciparum*. In the rodent malaria species *P. berghei*, Centrin4 was shown by super resolution structured illumination microscopy (SIM) to dynamically change its localization, from cytoplasmic outside of mitosis, to the CP during nuclear division [51]. In addition, PbCen4 was shown to associate with all other Centrins (PbCen1, 2 and 3); however, its role remains unclear, as its genetic abrogation had no effect on parasite cell division [51].

Akin to *Plasmodium* species, *T. gondii* asexual stages bear two distinct MTOCs; the APR and the centrosome [63]. The APR nucleates the subpellicular microtubules responsible for the parasite shape and motility. The centrosome nucleates the spindle microtubules crucial for mitosis [17,19,20,21,22,23,24]. The *T. gondii* centrosome is formed by two centrioles. Centrioles in this species are parallel to each other and are much shorter than their animal counterparts; centrioles in *T. gondii* are at the limit of optical resolution measuring approximately 250 nm length and width [20,33,64] (Figure 2B). The *T. gondii* centrioles display the characteristic centriolar nine-fold radial symmetry but are composed of single microtubules and a central tube. Single microtubule centrioles are rather rare in nature, but *T. gondii* is not exclusive in this respect. Centrioles of one-cell embryos of *Caenorhabditis elegans* display a comparable morphology [9,65]. However, single microtubule centrioles are associated with a lack of delta- and epsilon-tubulin coding genes in this species [66]. Conspicuously, both tubulin family members are easily identifiable in the *T. gondii* genome [67] but are presumably not expressed in asexual stages as they are indetectable in transcriptomic and proteomic analyses [67].

Pioneer transmission electron microscopy experiments dating to the early 60s identified an electron dense structure in asexually dividing tachyzoites which was first called the E-body; “E” referred to the endodyogeny mode of division used by the parasite [68]. Due to its high electron density, the E-body was initially thought to contain DNA. As the E-body changed morphology during division, the authors proposed this to be the factor which could “instigate” endodyogeny [68]. Dubremetz, Kalley, and Hammond recoined the structure under the term “centrocone,” as it is referred to nowadays [64]. The E-body or centrocone is a conical elaboration outlined by an outwards folding of the nuclear envelope (Figure 1C and Figure 2A). It is always positioned adjacent to the centrosome. Microtubules of the mitotic spindle are housed within the centrocone during mitosis penetrating the nuclear envelope through pores, contacting the kinetochores. This structure was later shown to be conserved in other related apicomplexans, such as *Eimeria* [64,69].

A protein carrying multiple membrane occupation and recognition nexus (MORN) motifs was shown to localize to ring structures at the apical and posterior end of the parasites, and to the centrocone [70]. This marker has allowed dynamic visualization of the structure, defining that the structure persists, but varies, throughout the cell cycle. At the start of mitosis, when centrosomes have not detectably separated yet, the centrocone noticeably protrudes from the nuclear envelope. It then duplicates and segregates along with the centrosome to two distinct, albeit adjacent, sites of the nucleus.

Akin to what was previously described for *Plasmodium*, the centrosome is apposed to a nuclear pore (Figure 1C). Centromeres of *T. gondii* cluster at the nuclear periphery permanently, at a location intimately related to that of the centrosome. MTs of the mitotic spindle are not present outside of mitosis. Centromere sequestration to the nuclear envelope in *T. gondii* is mediated by peripherally associated components of the nuclear pore complex [71].

In consonance for what has been described for *Plasmodium*, the *T. gondii* genome lacks homologs to many well-known centrosomal proteins. Protein coding genes for Spindle and centriole-associated protein 1 (SPICE), CEP135, CEP192, CEP63, CEP152, Spindle assembly abnormal protein 5 (SAS-5), CP110, Centrobin, POC5, C2CD3, Ofd1, Polo-like kinase -1 (PlK-1), and PlK-4 are absent from the genome (Table 1) [52,57,67,72]. Nonetheless, reciprocal BLAST searches in the genome, using the human centrosomal components as reference, have identified a number of relatively well conserved homologs. For example, homologs of SAS6, Centrins 1 thru 4, Centrin binding protein (Sfi1), and CEP250 have been not only identified in silico, but also validated as expressed proteins with centrosomal localization in *T. gondii* (Table 1).

As outlined above, SAS6 is a widely conserved centrosomal protein involved in ascribing centrioles with their characteristic morphology by means of assembling the centriolar cartwheel [3,4,5,6,7,8,9]. *T. gondii* bears two homologs of SAS6; TgSAS6 and TgSAS6-Like [49]. Interestingly though they are both located at MTOCs in *T. gondii’s* asexual stages, TgSAS6 is a centrosomal protein while TgSAS6L locates close to the APR [49]. Conspicuously, observation of a canonical cartwheel at the proximal end of centrioles has not been reported in *T. gondii*. The functional role of TgSAS6 at the centrosome remains unexplored. TgSAS6L is not essential for parasite survival. However, its characterization led the authors to propose that the apical MTOC likely evolved from a flagellar nucleating BB [49].This model is further supported by additional lines of evidence; for example, *T. gondii* has repurposed the algal derived striated fiber assembling (SFAs) to connect the centrosome to the apical MTOC. SFA proteins normally function anchoring BBs [73], but in T. gondii they serve to position the daughter cell’s APR during division [38].

Three centrin homologs have been identified and characterized in T. gondii; all of which localize to the centrosome [74]. In addition, Centrin2 localizes to the APR and to the parasite’s basal end. Centrin3 localizes also to the conoid [74] BLAST searches for additional Centrin homologs identify an additional homolog, bearing the most similarity to PfCen4 in *T. gondii* (TgME49_237490) [57] Though no expression of the putative TgCen4 is detectable in the asexual life forms of T. gondii, high levels of the transcript are detectable in sexual stages [75]. This expression pattern could reflect the involvement of the putative TgCen4 in BB biology. BBs may be either “recycled” or de novo assembled in male gametes of *T. gondii* [26]. It is plausible that either of these processes requires use of a different/specialized protein set than those found in asexual centrosomes. However, this hypothesis awaits experimental validation.

Centrin-binding proteins—Sfi—are yeast SPB proteins important for its replication [76]. In *T. gondii*, Suvorova and collaborators identified a centrin-binding protein homolog denominated TgSfi1 [52]. TgSfi1 localizes at the centrosome next to TgCentrin1 [52]. Interestingly this protein does not have a homolog in *Plasmodium*. However, a protein bearing a centrin binding motif has been identified. It has been proposed that the latter might play an akin function to that of TgSfi1, despite its lack of sequence conservation [52].

Centrosomal associated protein 250, also known as Cep250 or c-Nap1 (for centrosomal Nek associated protein), plays a role in maintaining centriole junction within the centrosome until the onset of mitosis in animals. A complex interplay of phosphorylation and dephosphorylation of Cep250 either prevents or triggers centriole disjunction in a cell-cycle dependent fashion [77]. Characteristically, human Cep250 bears four coiled-coiled domains. In *T. gondii*, Suvorova and collaborators identified TgCep250 (TGME49_212880) as a protein bearing seven predicted coiled-coiled regions—two of which display high similarity to that of HsCep250. Additionally, TgCep250-Like-1 was identified (TgCep250L1; TGME49_290620). This protein bears a single coiled-coiled domain but bears little homology to either HsCep250 or TgCep250 beyond this domain [52].

Fine localization by super-resolution microscopy of multiple protein components of the centrosome led to the proposition of a bi-modal *T. gondii* centrosome organization [52]. An outer core and an inner core were defined based on two distinct protein localizations with respect to the nucleus. The outer core faces the cytosol, while the inner core faces the nucleus. The outer core bears TgSas6, TgCentrin1, and TgSfi1. Hence, this core is thought to contain the atypical pair of centrioles described in T. gondii. However, this has not been formally shown. TgCep250 has been shown to localize to both the outer and inner cores, while TgCep250L1 localizes exclusively to the inner core. In addition, the centrosomal protein CEP530 has been recently shown to localize “in between” cores defining then a third location within the centrosome [78].

Functional insight on the role of the outer core has come from analyzing the functions of both TgCep250 and TgSfi1. TgCep250 has been described as required for keeping the outer core and the inner core connected during centrosomal replication [54]. Mutants of this protein display a dysregulation of inner and outer core replication, whereby the outer core over duplicates with respect to the inner core. Parasites displaying this phenotype fail to properly segregate the nucleus and lose the synchrony between nuclear mitosis and daughter cell formation [54]. Conversely, a temperature sensitive mutant bearing a non-synonymous point mutation in the TgSfi1 coding gene over-duplicates the inner core of the centrosome, whilst the outer core remains unduplicated. Consequently, this mutant exhibits profound cell division defects and fails to properly segregate its chromatin [52]. In both mutants, daughter cell assembly is severely impaired, displaying a characteristic drop in the number of daughter cells formed, and reinforcing the notion that outer core proteins play a pivotal role in orchestrating daughter cell formation.

Much less is understood about the inner core. This core has only been shown to house TgCep250 and TgCep250L1. The function of Cep250L1 has not been deciphered, hence, the precise role of the inner core remains ill-understood.

The mechanisms of MT nucleation by the *T. gondii* centrosome are not well understood. Characteristic electron density, corresponding to the PCM surrounding the centrioles, is not observable by electron microscopy in *T. gondii*. Consistently, homologs to many of the defining PCM core proteins in animals, such CEP192, pericentrin, and CDK5RAP2 are seemingly absent from the *T. gondii* genome [57] (Table 1). However, it should be noted that these searches are at best limited by our anthropocentric approach to the question; even many of the homologs that we do find are only distantly related to their animal counterparts.

Although the molecular players governing MT nucleation and spindle formation remain poorly defined, an intranuclear spindle has been clearly shown to be nucleated from the area surrounding the centrosome. Spindle microtubules are clearly observable within the centrocone (Figure 1C) [35,71,79,80]. This intranuclear spindle follows a cell cycle pattern originating from the centrosome and growing towards the nucleus. The first step in the mitotic spindle formation entails recruitment of tubulin to the centrocone [53,70]. This recruitment takes place at the end of G1 prior to the centrosome duplication [53]. As the cell cycle progresses, the spindle grows reaching the chromosomes [33,71,81].

Recent work identified a homolog of the microtubule plus end–binding protein EB1, known for its function as a mitotic spindle microtubule stabilizing factor in many systems [82]. TgEB1 has further allowed visualization of the dynamics of the mitotic spindle. TgEB1 localizes to the nucleus outside of division and quickly re-localizes to the centrocone region upon entry into M phase, coinciding with the time of spindle formation [53]. Mutants of this protein display lagging chromosomes, reinforcing the notion that a healthy spindle is required for proper chromosome segregation in these parasites.

On the other hand, though PCM components are missing, proteins known to play pivotal roles in centrosomal MT nucleation have been identified. Importantly, Ɣ-tubulin has been localized to centrioles, and more recently, finely mapped to the outer core [52,67]. In most organisms, MT nucleation is carried out by Ɣ-tubulin complexes, or Ɣ-TuSC. In animals, multiple Ɣ-TuSCs assemble with other proteins into Ɣ-tubulin ring complexes (Ɣ-TuRCs) [83]. However, Ɣ-Tusc have been shown to suffice for MT nucleation in some species [84]. The *T. gondii* genome encodes for GCP4, but not for GCP5 and GCP6 of the Ɣ-TuRC ring complex [57]. It is possible that MT nucleation could be orchestrated by the assembly of Ɣ-TusC ring instead of Ɣ-TuRC ring [57].

## 3. Nuclear Division Organization by MTOCs of *T. gondii* and *Plasmodium*: Insight into Regulatory Networks

Understanding the mechanism involved in the replication of the MTOC are crucial to coming closer to the molecular clues behind the speedy and flexible cell division of apicomplexan parasites.

In animal cells, new centriole biogenesis occurs at the onset of S-phase. The process is catalyzed by the phosphorylation of STIL by a polo-like kinase known as PLK4 [85]. This phosphorylation is “permitted” only upon S-phase entry when an inhibitory phosphorylation by CDK1/CyclinB is relieved in STIL [86]. Phosphorylated STIL can recruit SAS6 to the mother centriole wall to initiate new cartwheel formation. The interaction between SAS6 and CEP135 allows cartwheel stabilization and interaction with MTs which will form the centriole barrels [87].

Once the centrosome has replicated, pre-existing centrioles remain attached to each other through a physical linker formed by a multi-protein complex (involving CEP250/C-nap1 and rootletin). The dissolution of this link is critical to allow semi-conservative centrosome segregation to opposing poles, whereby each centrosome will be formed by a mother (old) and a (new) daughter centriole. Cep250 is the substrate of Nek2A, a kinase that belongs to the group of NIMA-related kinases family. In mammalian cells, Nek2A is substrate of an additional kinase called Mst-2, and of Protein Phosphatase 1 (PP1). In addition, Mst2’s activity is positively regulated by PLK-1. Phosphorylated Mst-2 phosphorylates Nek2A, which subsequently phosphorylates Cep250 inducing centrosome disjunction. Nek2A is dephosphorylated by PP1, reversing the process.

In *P. falciparum*, four Neks have been identified. PfNek-2 and PfNek-4 were shown to be essential for sexual development in the mosquito vector and consistently, are only expressed in the gametocyte stage. PfNek-1 is expressed both in male gametocytes and during asexual stages [88,89,90]. Interestingly, observation of PfNek1′s localization by immunofluorescence assays in asexually replicating *P. falciparum* revealed that it localizes to dots near the nucleus at the ring and schizont stages, but switches to a more diffuse cytoplasmic localization in trophozoites [88] PfNek-1 was shown to in vitro phosphorylate Pfmap-2, an atypical *P. falciparum* MAPK homolog [90].

Conversely, seven Neks have been identified in *T. gondii* [91]. TgNek-2 thru 7 remain unexplored. However, TgNek-1 was shown to localize at the centrosome at G1/S; the time of centrosome duplication in *T. gondii* [56]. In accordance with its predicted role, a mutant of TgNek-1 exhibits a single centrosome in asexually dividing parasites. This presumably corresponds to a centrosome whose disjunction is inhibited [56]. Interestingly, the CEP250 homologs (TgCEP250) is not a substrate of TgNek-1 [54]. Given TgNek-1′s localization at the outer core, and TgCEP250L1 localization to the inner core, TgCEP250L1 is likely not a substrate of TgNek-1 either. Analysis mapping phosphoproteome changes in the background of a TgNek-1 mutant could be informative as to the underlying cell-cycle regulated mechanisms of centriole junction and disjunction in *T. gondii*.

A single PP1 homolog has been identified in each *T. gondii*, *P. berghei*, and *P. falciparum* [92,93,94,95,96,97]. Given their critical role in controlling proliferation in other organisms PP1 homologs in apicomplexans have been put forward as promising drug targets to block parasite division.

The PP1 holoenzyme is conformed by a catalytic subunit highly conserved denominated protein phosphatase 1 catalytic subunit (PP1c), which forms a complex with different regulatory subunits. The diversity of regulatory subunits that modulates PP1c phosphatase activity is such that a “binding code” for PP1c has been proposed, whereby the different combination of these subunits creates holoenzymes with unique properties [98]. In fact, most of the PP1c surfaces are interaction interfaces with different proteins. This, in turn, limits PP1c evolutionary rate, making it one of the most highly conserved enzymes amongst eukaryotic lineages [99]. It should be noted that PP1 plays numerous roles other than limiting centrosome separation in many organisms, however, discussing those extends well beyond the scope of this review.

Interactions of PP1c with different players control its spatio-temporal activity. One such interaction, relevant to centrosome disjunction, is that of PP1c with its specific inhibitor. The “Inhibitor 2” (I-2) is a cell-cycle regulated PP1c inhibitor which is specifically expressed in S and M phases. In animal cells, this protein localization to the pericentriolar area, coincides with an increase in the kinase activity of the Nek2A-Mst2-PP1 complex (i.e., an inhibition of the phosphatase activity of PP1) [100]. An I-2 homolog has been shown to exist in *T. gondii* and is named TgI2. TgI2 was shown to inhibit TgPP1′s phosphatase activity in vitro. This inhibition is critically dependent on TgI2′s SILK and RVxF motifs, a feature conserved in the higher eukaryotes I-2s [101]. In addition, A leucine-rich repeat protein family, TgLRR1, binds TgPP1 within the nucleus. This interaction, assayed using recombinant proteins, was shown to inhibit TgPP1′s phosphatase activity in Xenopus oocytes, overriding the G2/M cell cycle checkpoint in this system [102]. Overall, TgPP1 is predicted to play a pivotal role in controlling cell cycle progression, likely through a prominent role in centrosome duplication. Though its phosphatase activity has been shown to be critically dependent on TgPP1′s interactions with its specific inhibitor TgI2, and its binding partner TgLRR1 (which likely limits its activity at the centrosome by compartmentalizing it to the nucleus), nothing is known about its substrates nor about its in vivo interaction with TgNek1. Elucidating these critical aspects of TgPP1′s life could shed light onto ill-understood, yet critical, aspects of centrosome biology in *T. gondii*.

In *P. berghei*, live-cell and ultrastructural imaging, showed recently that PbPP1 cyclically localizes to the proximity of the nucleus, at a position apposed to that of NDC80, a marker of *Plasmodium*’s kinetochores [55]. This localization coincides with the start of DNA synthesis/S-phase. Immunofluorescence assays revealed similar localization dynamics of the PP1 homolog in P. falciparum whereby a diffuse cytoplasmic and more intense foci near the nucleus could be observed [103]. Interestingly, both PfLRR1 and PfI-2 homologs have been identified, suggesting that the players involved in regulating PP1′s activity at the centrosome in higher eukaryotes could be conserved in *Plasmodium* [104]. However, though a number of studies have focused on shedding light onto PP1′s functions in gametogenesis, egress and host–parasite interactions [105], through conditional mutagenesis, RNA-seq and proteomics, whether PP1 plays any role in *Plasmodium*’s CP biology remains to be determined.

The coordinated and timely onset of successive cell cycle stages is largely controlled by Cyclin-dependent kinases (CDKs) in mammalian cells. Many of the CDKs control cell cycle progression by means of what are known as “checkpoints”. Bona fide cell cycle progression check points—as defined by the stalling of one process when another one has not progressed properly—are seemingly absent in *T. gondii*. This phenomenon has been repeatedly documented by phenotypic characterization of cell division mutants. An illustrative example is the mutant of the kinetochore protein TgNdc80. TgNdc80 conditional knock-down parasites lose the connection between the nucleus and the centrosome. In these mutants, the nucleus “falls off” the mother cell, whilst it continues on with daughter cell assembly [106]. On the flip side, mutants who lose the connection between the centrosome and the MTOC guiding daughter cell assembly, are able to undergo mitosis normally [38]. Instead of checkpoints, temporally coinciding mutual physical tethers are assembled onto the centrosome. Proper spatial and temporal co-organization of cell division events is ensured in this fashion [38,54].

Nonetheless, a number of Cdk-related kinases (Crks; TPK2, TgCrk1, TgCrk2, TgCrk4, TgCrk5, and TgCrk6) and in some cases their partner cyclins, have been identified and characterized in *T. gondii* [107,108,109]. TgCrk6 and TgCrk4 are required for progression through S-phase and mitosis. These CRKs have been proposed to partake in the regulation of spindle assembly and centrosome duplication, respectively. Their cyclin partners have not been identified, nor have their substrates been deciphered.

Seven CRKs have been identified in *Plasmodium* (PK5, PK6, Mrk1, Crk-1, Crk-3, Crk-5, and Crk-4) [110,111], as well as four cyclins (Cyc1, Soc2, Cyc3, and Cyc4) [112,113]. CRK5 interacts with cyclin SOC2. Together, these proteins have been shown to play a role in licensing DNA replication. Consistently, a *P. berghei* mutant of CRK5 exhibits fewer nuclear poles, no chromatin condensation, fails to undergo cytokinesis or form flagella [114]. PfPK6 is also proposed to regulate S phase entry, however, its precise function and its interactors remain unknown [115]. To our knowledge, which CRK/Cyclin pairs are specifically involved in catalyzing spindle formation or CP duplication in *Plasmodium* have not been identified.

Mitogen-activated protein kinases (MAPKs) are a conserved family of protein kinases that regulate signal transduction, proliferation, and development in eukaryotes. The genome of *T. gondii* encodes for three MAP-related kinases; MAPKL1, MAPK2, and ERK7. ERK7 has been shown to be involved in APR homeostasis and biogenesis [116]. Conditional null parasites for this protein exhibit a striking phenotype whereby conoid assembly is completely abrogated. On the other hand, both MAPKL1 and MAPK2 have been implicated in the regulation of centrosome duplication. A temperature sensitive mutant of MAPKL1 over-duplicates the centrosome at restrictive temperature, leading to the assembly of an aberrant number of daughter cells [52]. TgMAPKL1′s substrates, however, remain unidentified. On the other hand, conditional degradation of MAPK2 renders parasites unable to duplicate their centrosomes, complete DNA replication, and initiate daughter cell budding. However, prior to succumbing, MAPK2 mutant parasites continue to grow and replicate their mitochondria, Golgi apparatus and plastid-like organelle, the apicoplast [117]. The latter two are known to segregate with the centrosome. Nonetheless, MAPK2 does not localize at the centrosome, for which its function is likely exerted upstream of centrosome duplication and mitosis.

Aurora-related kinases are a family of serine/threonine kinases well known for critically regulating cell cycle progression in many organisms. *P. falciparum* and *T. gondii* each bear three homologs of these kinases [118,119]. In *P. falciparum* these kinases are denominated: Pfark-1,-2,-3 and in *T. gondii*: TgArk-1,-2,3 [118,120]. *T. gondii*’s Ark1-3 have been experimentally addressed and all are functionally related to licensing events of the cell cycle [118,121]. TgArk-1 has been proposed to play a role in the duplication of the spindle pole and the inner core of the centrosome [121]. TgArk-2 is the only of three that is not essential for tachyzoite replication [118]. Although it localizes at the intranuclear mitotic spindle, mutants of TgArk-2 are able to proliferate normally [118]. TgArk-3 localizes at the outer core of the centrosome and has been linked with regulation of the budding process, as mutants for this protein fail to assemble daughter cells and complete cytokinesis [52,121].

Three ARK homologs have been identified in *P. falciparum*; Pfark-1, -2, -3. Of these, Pfark1 has been experimentally explored. Pfark-1 has been shown to be essential in blood stages for parasite’s survival, and it has been shown to functionally interact with PfNek-1 [120,122]. PfArk-1 localizes at the spindle poles during mitosis. In metazoans, Aurora A localizes at the spindle during nuclear division [120]. Given its cell cycle stage dependent localization to the spindle, PfArk-1 has been proposed as the functional homolog of Aurora A [120]. Much less is known about PfArk-2: where it localizes, whether it follows a cell cycle-dependent pattern of localization, and its functions/substrates, remain undetermined [120]. In vitro assays showed PfArk-2 preference for myelin basic protein [123]. Finally, PfArk-3 displays a perinuclear localization and is expressed at the onset of S phase [123]. All three PfArks are essential for blood stage parasite survival [123].

## 4. Closing Remarks

Though MTOC structures in Apicomplexa have captivated the interests of electron microscopists since the 60s, several limitations have precluded the identification of their molecular makeup. Mechanistic studies have been limited both by the minute sizes of the structures, and the difficulty in their purification: a characteristic inherent to their multiple connections to rather stable and resistant cytoskeletal structures. This has precluded the extended use of proteomic-based approaches for identification of centrosomal/CP proteins in either species.

In addition, the fast cell cycles of apicomplexan parasites preclude the detailed study of the various short-lived cell cycle stages of asexual proliferation, and the transitions between asexual and sexual life forms. For example, in asexually replicating *T. gondii* and *Plasmodium*, centrosome/CP duplication occurs at the onset of S-phase; a stage reckoned to last about half an hour in a 6 h cell division cycle. In asynchronous growing cultures of *T. gondii*, only a minute fraction of all parasites will be at this stage, making the study of the process a literal “needle in a haystack” kind of a challenge.

Cell cycle synchronization tools have tremendously increased our capacity to discern the events taking place in the mitosis of human cells. Synchronization tools based on differential osmotic stress of schizont stages are available for *Plasmodium* blood-stages. These tools, however, only enrich for ring stages and scalability of in vitro cultures remains a challenge. Cell cycle synchronization tools, which do not significantly modify the biology of the parasite and maintain synchrony for a significant period, have not been reported for use in *T. gondii*.

The puzzle of the various regulatory networks linking the different synchronous events of cell division in apicomplexans, has only recently started to come together. However, again, the various life forms and complex regulatory networks operating simultaneously in asynchronously growing parasites, transitioning between asexual and sexual cycles, makes the puzzle seem like an unapproachable challenge.

However, many of these limitations began to be relieved with the widespread access to efficient mutagenesis tools, super-resolution optical microscopy, and the application of extremely high-resolution electron microscopy techniques (such as Cryo-electron tomography and Correlative Light EM). More recently, the advent of UExM has democratized the access to higher resolution optical microscopy and fluorescence imaging [124,125]. The latter will most likely become the technique of choice in less well funded settings where access to state of the art super resolution microscopy is limited. Together, these techniques have expedited progress in our molecular understanding of MTOCs and their associated structures in the last few years, particularly impacting our understanding of *Plasmodium*’s CP role in MT nucleation and *T. gondii*’s centrosome organization. In addition, tools to in vitro trigger the development of sexual stages of *T. gondii*, allowing unprecedented access to flagella-forming stages (i.e., basal body biology!), are now available [30,31].

We envision that the next few years will see multiple breakthrough studies solving the fascinating puzzle of MTOC biology in *T. gondii* and *Plasmodium*, bringing us closer to both understanding the intricacies of these parasites’ basic biology, and devising new strategies to interfere with their most destructive power; their ability to proliferate within us.

## Figures and Tables

**Figure 1 microorganisms-09-02503-f001:**
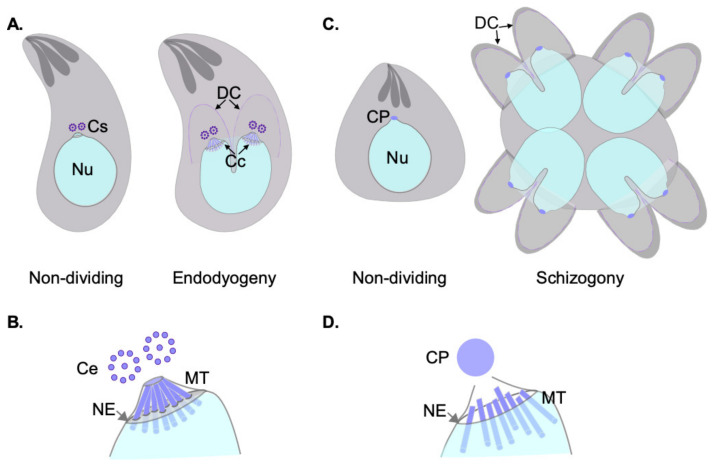
Cell division modes and nuclear MTOCs of asexually dividing *Toxoplasma gondii* and *Plasmodium* parasites. (**A**) Schematic representation of a *T. gondii*’s tachyzoites non-dividing and going through its cell division mechanism of endodyogeny, as indicated. Note that the intra-nuclear spindle is only assembled during division, while an elaboration of the nuclear envelope (marking the site of centrocone protrusion during cell division) is observable in non-dividing parasites. Ce; centrioles, are shown in purple. Nu; nucleus, Cs; centrosome, DC; daughter cells, Cc; centrocone. (**B**) Schematic representation of the centriole morphology, and the intra-nuclear spindle formed during *T. gondii* division. Note that microtubules (MT) are nucleated within the centrocone, a structure contained within the nuclear envelope (NE). MTs go through pores of the NE. (**C**) Schematic representation of asexual stage of *Plasmodium* spp. both non-dividing and dividing by schizogony, as indicated. Note the centriolar plaque (CP), the nuclear MTOC lies in the proximity of the nucleus in all stages. (**D**) Schematic representation of the centriolar plaque (CP). Microtubules (MTs) are nucleated during division in *Plasmodium* spp. at a region within the nuclear envelope (NE) devoid of chromatin, and physically distinct from the CP which lies outside the nucleus.

**Figure 2 microorganisms-09-02503-f002:**
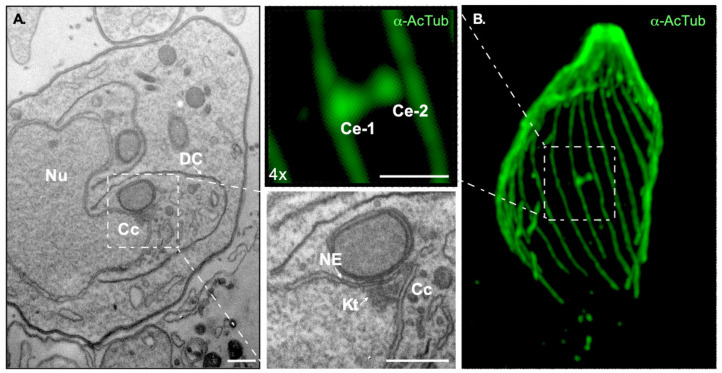
High resolution images of MTOC associated structures in *Toxoplasma gondii*. (**A**) Transmission electron micrograph of dividing *T. gondii*’s tachyzoites by endodyogeny. The centrocone (Cc), the dividing nucleus (Nu) in the process of segregation into two daughter cells (DC) are observable. The inset shows a detailed overview of the nuclear envelope (NE) forming the centrocone (Cc) and the kinetochores (Kt). (**B**) Confocal microscopy image of ultrastructure-expanded non-dividing *T. gondii*’s tachyzoite labeled with anti-acetylated Tubulin, as indicated. Note that using this technique the pair of centrioles (Ce1 and Ce2) forming the centrosome in this parasite are observable and can be resolved. Scale = 500 nm in all cases.

**Table 1 microorganisms-09-02503-t001:** *Toxoplasma gondii* and *Plasmodium falciparum* homologs of mammalian centrosomal proteins.

Gene ID	*T. gondii* Gene ID (TGME49_)	*P. falciparum* Gene ID (Pf3D7)	Role in *T. gondii* Survival	Role in *Plasmodium* Survival
SAS-4/C-PAP	258710	1458500	Essential by HTF	Not essential by HTS
CEP120	285210	-	Not essential by HTS	-
CEP76	226610	-	Not essential by HTS	-
POC1	216880	0826700	Essential by HTS	Essential by HTS
SAS6	306430	0607600	Not essential by HTS	Not essential by HTS
SAS6L	301420	1316400	Essential by SGKO [49]	Not essential by HTS
CEP135	-	0626500	-	Not essential by HTS
Centrin 1	247230	0107000	Essential by HTS	Not data available
Centrin 2	250340	1446600	Likely Essential by SGKO [50]	Not essential by HTS
Centrin 3	260670	1027700	Essential by HTS	Not essential by HTS
Centrin 4	237490	1105500	Not essential by HTS	Not essential by SGKO [51]
Sfi1	274000	-	Essential by SGKO [52]	-
CEP164	314358	-	Essential by HTS	-
CEP170	201790	1307800	Essential by HTS	Not essential by HTS
CEP110	211430	1032800	Not essential by HTS	Essential by HTS
kif24	287160	1245100	Not essential by HTS	Not essential by HTS
EB1	227650	0307300	Not essential by SGKO [53]	Not essential by HTS
CEP250	212880	-	Essential by SGKO [54]	-
CEP250L1	290620	-	Essential by HTS	-
PP1	310700	1414400	Essential by HTS	Essential by SGKO [55]
Nek2/NimA	292140	1228300	Essential by SGKO [56]	Not essential by HTS
LLRC45	209830	-	Not essential by HTS	-
CEP72	233940	1347800	Not essential by HTS	Essential by HTS
CEP131	205590	-	Not essential by HTS	-

Adapted from [57]. HTS: high throughput CRISPR-Cas9 mutagenesis-based screening; SGKO: single gene knockout.

## Data Availability

Not applicable.

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
