# Peer review of "Structural and Functional Insights into the Microtubule Organizing Centers of Toxoplasma gondii and Plasmodium spp."

_microorganisms, 2021, doi:10.3390/microorganisms9122503_

Round 1

Reviewer 1 Report

In the review entitled “Structural and functional insights into the microtubule organizing centers (MTOCs) of Toxoplasma gondii and Plasmodium spp.”, the authors give an overview of MTOCs in the different stages of Toxoplasma and Plasmodium spp  and compare with what is known for animal and yeast cells. Overall, the review is informative and interesting. The concluding remarks touch on the complications inherent in the topic and provide food for thought for future work.

Major points:

  • In Figure 1B, please list what Ce stands for in the legend.
  • In Figure 1B, it would be helpful to add arrows to point out the NE (The section that has pores that the MT go through and the NE containing the centrocone)
  • On line 248, the authors state that centromeres dissociate soon after invasion. In which stage of the parasite is this or is it all stages? If the centrosomes cluster to a single nuclear location before mitosis, when do they reform after invasion?
  • On lines 608-611, the authors state that in an MAPK2 Toxoplasma mutant, the parasites do not initiate daughter cell budding (etc) but continue to grow. Is this mutant capable of propagating even though there is a defect in daughter cell budding or was a conditional mutant used in the study?

Minor points:

  • On line 178, using the term “if regarded” in the sentence seems (upon a first read) to limit the diversity of centrosomal architecture between apicomplexan and animals but there is also diversity amongst apicomplexan parasites (as stated in the next sentence). It would be clearer to omit “if regarded”.
  • On line 206, switch the ; for : (since a list of items follows). Plus, it would be easier to read if the “and” between “CPs” and “the ones” is changed to a comma

Author Response

Dear Reviewer 1

We would like to thank you for taking the time to critically review our manuscript entitled “Structural and functional insights into the microtubule organizing centers (MTOCs) of Toxoplasma gondii and Plasmodium spp.”, We appreciate  that you reckon our review informative and interesting, and that you found our concluding remarks provide food for thought for future work. Below please find a point by point response (in red) to the major and minor points you raised regarding our manuscript. We appreciate your suggestions which have improved the clarity and quality of our manuscript, and hope that you now find it suitable for publication in Microorganisms.

Reviewer 2 Report

It is an up-to-date review on the microtubule organizing centers of Apicomplexan parasites.

Minor suggestions:

Use italic letters for species and genus names.

Do not use the full name of the species; only in the first mention. E.g., T. gondii instead of Toxoplasma gondii.

Some abbreviations (MT, NE) are not applied consequently instead of the whole words (microtubule, nuclear envelope) are used.

Chapter 1, line 107: Write single flagellum instead of single flagella.

Chapter 2: In the title write MTOCs instead of mtocs.

Line 204. Maybe TEM -transmission electron microscopy - could be resolved.

Line 235: Similarly, STED can be resolved - stimulated emission depletion.

Line 364: Use homologs instead of homologes.

Line 387: Use homolog and homologs instead of homologue and homologues, respectively. (It seems that the authors use American spelling otherwise. Use consequently British or American spelling. E.g., line 459 homolog and line 358 homologs, but line 447 homologues.)

Table 1

The last column cannot be fully seen.

Chapter 3: Do not use capital letters in the title.

References

Use italic letters for species and genus names.

Ref 8: Journal name should be italic not the title.

Ref 20: J Cell Sci 126(Pt 15):3344-3355.

Ref 35: J Microsc. - Full journal title: Journal of Microscopy

References 45, 52, 64, 70, 107: delete “PubMed” from these references.

Ref 55: Do not use capital letters in the title.

Ref 90: journal name must be italic.

Ref 91: Parasitology, 106 (Pt 3):223-232.

Ref 100: Full journal title: J Ultrastruct Res.

Journal of Ultrastructure Research, 59(3):332-350.

Author Response

Dear Reviewer 2

We would like to thank you for taking the time to critically review our manuscript entitled “Structural and functional insights into the microtubule organizing centers of Toxoplasma gondii and Plasmodium spp.”, Below please find a point by point response (in red) to the minor points you raised regarding our manuscript. We appreciate your suggestions which have improved the clarity and quality of our manuscript, and hope that you now find it suitable for publication in Microorganisms.
